# Psoriatic Resolved Skin Epidermal Keratinocytes Retain Disease-Residual Transcriptomic and Epigenomic Profiles

**DOI:** 10.3390/ijms24054556

**Published:** 2023-02-25

**Authors:** Ameneh Ghaffarinia, Ferhan Ayaydin, Szilárd Póliska, Máté Manczinger, Beáta Szilvia Bolla, Lili Borbála Flink, Fanni Balogh, Zoltán Veréb, Renáta Bozó, Kornélia Szabó, Zsuzsanna Bata-Csörgő, Lajos Kemény

**Affiliations:** 1HCEMM-USZ Skin Research Group, H-6720 Szeged, Hungary; 2Department of Dermatology and Allergology, Albert Szent-Györgyi Medical School, University of Szeged, H-6720 Szeged, Hungary; 3HCEMM-USZ, Functional Cell Biology and Immunology, Advanced Core Facility, H-6728 Szeged, Hungary; 4Laboratory of Cellular Imaging, Biological Research Centre, Eötvös Loránd Research Network, H-6726 Szeged, Hungary; 5Institute of Plant Biology, Biological Research Centre, H-6726 Szeged, Hungary; 6Genomic Medicine and Bioinformatics Core Facility, Department of Biochemistry and Molecular Biology, Faculty of Medicine, University of Debrecen, H-4032 Debrecen, Hungary; 7Systems Immunology Research Group, Institute of Biochemistry, Biological Research Centre, ELKH, H-6726 Szeged, Hungary; 8HCEMM-Systems Immunology Research Group, H-6726 Szeged, Hungary; 9ELKH-SZTE Dermatological Research Group, Department of Dermatology and Allergology, University of Szeged, H-6720 Szeged, Hungary; 10Regenerative Medicine and Cellular Pharmacology Laboratory (HECRIN), Department of Dermatology and Allergology, University of Szeged, H-6720 Szeged, Hungary; 11Research Institute of Translational Biomedicine, Department of Dermatology and Allergology, University of Szeged, H-6720 Szeged, Hungary

**Keywords:** psoriasis, keratinocyte, relapse, transcriptomics, epigenomics, 5-mC, 5-hmC

## Abstract

The disease-residual transcriptomic profile (DRTP) within psoriatic healed/resolved skin and epidermal tissue-resident memory T (TRM) cells have been proposed to be crucial for the recurrence of old lesions. However, it is unclear whether epidermal keratinocytes are involved in disease recurrence. There is increasing evidence regarding the importance of epigenetic mechanisms in the pathogenesis of psoriasis. Nonetheless, the epigenetic changes that contribute to the recurrence of psoriasis remain unknown. The aim of this study was to elucidate the role of keratinocytes in psoriasis relapse. The epigenetic marks 5-methylcytosine (5-mC) and 5-hydroxymethylcytosine (5-hmC) were visualized using immunofluorescence staining, and RNA sequencing was performed on paired never-lesional and resolved epidermal and dermal compartments of skin from psoriasis patients. We observed diminished 5-mC and 5-hmC amounts and decreased mRNA expression of the ten-eleven translocation (TET) 3 enzyme in the resolved epidermis. SAMHD1, C10orf99, and AKR1B10: the highly dysregulated genes in resolved epidermis are known to be associated with pathogenesis of psoriasis, and the DRTP was enriched in WNT, TNF, and mTOR signaling pathways. Our results suggest that epigenetic changes detected in epidermal keratinocytes of resolved skin may be responsible for the DRTP in the same regions. Thus, the DRTP of keratinocytes may contribute to site-specific local relapse.

## 1. Introduction

Psoriasis is a relapsing–remitting immune-mediated skin disorder characterized by epidermal hyperplasia, and massive inflammatory infiltrates as hallmarks of scaly erythematous lesions [1,2]. The global prevalence of psoriasis is around 125 million people worldwide [3]. Successful treatment of psoriasis leads to the normalization of epidermal thickness, reduced leukocyte infiltrate, and the return to a clinically and histologically normal condition [4,5,6], referred to as resolved skin. However, many patients do not attain sustained remission and often experience the reappearance of cutaneous symptoms within weeks or months of cessation of the therapy [7,8]. The psoriatic lesions frequently occur on the so-called predilection sites, such as knees and elbows, and they usually reappear on the same body parts. On the other hand, patients can also identify body regions never affected by psoriasis, which we can refer to as never-lesional skin. Previously, we showed that the psoriatic uninvolved skin in patients with severe plaque-type psoriasis maintains molecular, cellular, and extracellular alterations and displays a “pre-psoriatic” phenotype; however, in these studies, we did not discriminate between psoriatic never-lesional and resolved uninvolved skin [9,10,11,12,13]. Since disease flare-up most commonly occurs in the resolved skin, we decided to distinguish between resolved and never-lesional skin to find clues for the recurrence of the disease.

Tissue-resident memory (TRM) T cells retained in resolved skin have been suggested as the main drivers of psoriasis relapse [9,14,15,16]. However, whether structural skin cells such as epidermal keratinocytes play a role in the recurrence of the clinical symptoms or not is unclear. At the molecular level, previous bulk microarray and RNA sequencing studies have shown abnormal expression of immune and skin structural genes within resolved skin [14,15,16,17,18]. Nonetheless, it is difficult to dissect the cell-specific transcripts in a whole-skin transcriptome due to its diverse cell populations. Furthermore, it is now apparent that epigenetic regulatory mechanisms such as DNA methylation and DNA hydroxymethylation play a pivotal role in the pathogenesis of several skin diseases, such as psoriasis [19,20,21,22,23,24,25,26,27,28,29,30]. However, the epigenetic changes contributing to psoriasis relapse remain unknown. 

In order to clarify the role of keratinocytes in the recurrence of lesions in resolved skin, we looked at epidermal keratinocyte transcriptomic and epigenomic differences in resolved vs. never-lesional skin. Since the specific transcripts of epidermal keratinocytes often remain masked in whole-skin transcriptomic analyses, we performed high-throughput RNA sequencing separately on the epidermal and dermal compartments. The visualization of the general pattern of DNA methylation and DNA hydroxymethylation epigenetic marks, 5-mC (5-methylcytosine) and 5-hmC (5-hydroxymethylcytosine) was performed on the paired resolved and never-lesional skin sections using immunofluorescence staining. Finally, we examined the overlap between resolved vs. never-lesional skin in terms of differentially expressed genes (DEGs) from our study, and the lesional vs. healthy skin DEGs from the available datasets. 

We observed decreased contents of 5-mC and an apparent loss of 5-hmC in resolved epidermis vs. never-lesional epidermis. Additionally, the 5-hmC decreased levels were accompanied by decreased ten-eleven translocation (TET) 3 mRNA expression in resolved epidermis as compared to never-lesional epidermis. There were 102 genes that overlapped among the DEGs of resolved vs. never-lesional epidermis and lesional vs. healthy epidermis. These data suggest that epidermal keratinocytes of resolved skin may contribute to a local relapse in psoriasis, possibly because they are not epigenetically and transcriptionally fully recovered to the baseline level (never-lesional skin). The experimental workflow is summarized in Figure 1.

## 2. Results

### 2.1. 5-mC and 5-hmC Amounts Were Overall Lower in Resolved Epidermis as Compared to Never-Lesional Epidermis

Using immunofluorescence staining and confocal microscopy, we observed that 5-mC and 5-hmC amounts were overall lower in resolved epidermis as compared to never-lesional epidermis. We detected a uniform 5-mC distribution pattern in both never-lesional and resolved epidermis and enhanced 5-hmC intensity in the suprabasal layer of never-lesional vs. resolved epidermis (Figure 2a). On higher resolution images (oil immersion objective), 5-mC and 5-hmC nuclei staining were apparently weaker in the resolved epidermis compared to the never-lesional samples (Figure 2a, closeups). False-colored low-magnification images reported these findings in each patient, although a few 5-hmC hot patches were observed in the resolved epidermis (Figure 3).

### 2.2. 5-hmC Weaker Intensity Was Accompanied by Decreased TET3 mRNA Expression Level in Resolved Epidermis 

The 5-hmC epigenetic mark is an intermediate product of active DNA demethylation. This refers to an enzymatic process in which ten-eleven translocation (TET)1, TET2, and TET3 enzymes oxidize the methyl group of 5-mC and convert it to 5-hmC [31,32]. Therefore, we analyzed mRNA expression levels of these enzymes in paired never-lesional and resolved epidermal samples to verify whether there was a link between 5-hmC loss in resolved epidermis and gene expression alterations of these enzymes. The real-time RT-PCR results showed a significant decrease in TET3 (*p* = 0.0002) and no changes in TET1 and TET2 mRNA expression level in resolved vs. never-lesional epidermis (Figure 2b). Overall, we observed a correlation between a 5-hmC decreased level and TET3 mRNA expression in resolved epidermis. These results are consistent with previous findings confirming that the loss of 5-hmC was associated with lower mRNA expression of TET enzymes in the epidermis of psoriatic lesions [28]. 

### 2.3. Transcriptome Profiling of the Resolved Epidermis vs. Never-Lesional Epidermis

#### 2.3.1. Analysis of Differentially Expressed Genes (DEGs)

A pairwise comparison of RNA sequencing profiles from paired resolved and never-lesional epidermal and dermal compartments was performed to determine differences in gene expression. Transcriptome analysis of four patients samples yielded 476 DEGs in resolved epidermis vs. never-lesional epidermis (*p* < 0.05). Of these, 275 (≈57.77%) were down-regulated, and 201 (≈42.23%) were up-regulated. The same analysis of three patients yielded 2966 DEGs in resolved dermis vs. never-lesional dermis (*p* < 0.05). Of 2966 DEGs, 1360 (≈45.85%) were identified as down-regulated, and 1606 (≈54.15%) as up-regulated genes in the resolved dermis vs. never-lesional dermis. Principle component analysis (PCA) was performed using the resolved vs. never-lesional DEGs as input. The analyses showed a clear separation between all resolved and never-lesional paired samples (Appendix A, see Appendix A).

To narrow the search for candidate genes that may play a role in psoriatic lesion recurrence, we examined the 25 most down-regulated and up-regulated DEGs in resolved vs. never-lesional epidermal and dermal compartments (Table 1). 

#### 2.3.2. The Most Down-Regulated DEGs in Resolved Epidermis vs. Never-Lesional Epidermis (|FC| ≥ 2.5)

Overall, three genes with |FC| ≥ 2.5 were decreased in the resolved epidermis compared with the never-lesional epidermis, including nuclear enriched abundant transcript 1 conserved region 3 (**NEAT1_3**), sterile alpha motif and HD domain-containing protein 1 (**SAMHD1**), and Homeobox protein Hox-B2 (**HOXB2**) (Table 1). Among these, NEAT1_3 and SAMHD1 can act as inflammation regulatory genes. The NEAT1_3 gene produces a long non-coding RNA (lncRNA) with critical roles in innate immune responses [33,34,35]. Screening of the expression of some common lncRNAs in psoriasis using real-time RT-PCR has shown decreased expression of the NEAT1 gene in lesional skin compared to healthy skin [36]. The gene SAMHD1 is the only deoxynucleoside triphosphohydrolase (dNTPase) in eukaryotes, and is required for genome integrity by tightly controlling the intracellular deoxynucleoside triphosphate (dNTP) pools to prevent toxic dNTP accumulation [37]. Using RNA sequencing analyses, SAMHD1 was shown to be significantly up-regulated in psoriatic lesional skin compared to healthy skin (fold of change, 1.5) [38].

#### 2.3.3. The Most Up-Regulated DEGs in Resolved Epidermis vs. Never-Lesional Epidermis (|FC| ≥ 2.5)

A total of six genes, including SH3 and cysteine-rich domain 2 (**STAC2**), aquaporin 5 (**AQP5**), family with sequence similarity 25 member C (**FAM25C**), ELOVL fatty acid Elongase 3 **(ELOVL3**), chromosome 10 open reading frame 99 (**C10orf99**), and aldo-keto reductase family 1 member B10 (**AKR1B10**), were increased with a change greater than 2.5-fold in resolved epidermis vs. never-lesional epidermis (Table 1). Two of these transcripts, C10orf99 and AKR1B10, were highly expressed in the lesional skin of psoriasis patients compared to healthy controls [39,40,41]. AKR1B10 is a human NADPH-dependent oxidoreductase [42]. The AKR1B10 enzyme plays a role in lipid metabolism and functions as a positive regulator of inflammation. Inhibition of AKR1B10 can suppress the inflammatory response triggered by various stressors in cellular and animal models [43,44,45,46,47]. AKR1B10 plays a role in psoriasis lesion formation by dysregulating the retinoic acid signaling pathway, thereby inducing the excessive proliferation of keratinocytes [41]. The gene C10orf99 encodes the protein GPR15L, a novel antimicrobial peptide with broad activity against various bacteria and fungi [48]. GPR15L is mainly expressed in epithelial tissues and functions as a homeostatic chemokine for recruitment of mouse dendritic epidermal T-cell into skin [49,50,51].

#### 2.3.4. The Most Down-Regulated DEGs in Resolved Dermis vs. Never-Lesional Dermis (|FC| > 10)

Because all 25 most down-regulated and up-regulated DEGs in resolved dermis vs. never-lesional dermis had a change greater than 2.5-fold, we decided to increase the limit to ≥10 to restrict the search to the most down-regulated and up-regulated DEGs (Table 1). The most down-regulated genes in resolved dermis vs. never-lesional dermis included small proline-rich protein 4 (**SPRR4**), ATPase H+/K+ transporting non-gastric alpha2 subunit (**ATP12A**), cystatin-E/M (**CST6**), Small proline-rich protein 1A (**SPRR1A**), cysteine-rich C-Terminal 1 (**CRCT1**), microseminoprotein β (**MSMB**), keratin 34 (**KRT34**), and gap junction protein beta 4 (**GJB4**). To the best of our knowledge, only MSMB gene down-regulation has been shown to be associated with pathogenesis of psoriasis. MSMB has been previously shown to be down-regulated in the lesional skin of psoriasis patients compared to non-lesional and healthy skin [40,52,53]. 

#### 2.3.5. The Most Up-Regulated DEGs in Resolved Dermis vs. Never-Lesional Dermis (|FC| > 10)

The most up-regulated genes in resolved dermis vs. never-lesional dermis were immunoglobulin heavy variable 3-7 (**IGHV3-7**), immunoglobulin heavy variable 3-33 (**IGHV3-33**), and immunoglobulin heavy joining 4 (**IGHJ4**) (Table 1). Interestingly, of the 25 most up-regulated DEGs in resolved dermis, 14 transcripts, accounting for 56%, were immunoglobulin coding gene segments. The high prevalence of B cell receptor gene segments in resolved vs. never-lesional dermis raises the possibility that B cells may play a role in psoriasis local relapse. However, its importance is unknown and further studies are needed to investigate the contribution of B cells to the local recurrence of psoriatic lesions. 

#### 2.3.6. Identifying Biological Processes (BPs), Molecular Functions (MFs), and Cellular Components (CCs) Enriched among the DEGs

We performed gene ontology (GO) enrichment analysis for resolved vs. never-lesional DEGs. The results showed that 75 BPs, 32 MFs, and 7 CCs were significantly overrepresented in resolved vs. never-lesional epidermis (*p* < 0.05). In the GO analysis of the five most significant BPs, genes were mainly associated with the following terms: regulation of protein phosphorylation, regulation of phosphorylation, regulation of stress-activated MAPK cascade, positive regulation of stress-activated MAPK cascade and secondary heart field specification. In the five most significant MFs, genes were associated with the following terms: interleukin-6 receptor binding, death receptor activity, cysteine-type endopeptidase activity involved in the apoptotic process, positive regulation of telomerase activity, and regulation of cysteine-type endopeptidase activity. Most of these responses occurred in the early endosome lumen (Figure 4).

The results in the dermal samples revealed that 37 BPs, 6 MFs, and 29 CCs were significantly overrepresented in resolved vs. never-lesional (*p* < 0.05). In the five most significant BPs, genes were associated with the following terms: detection of chemical stimulus, detection of stimulus involved in sensory perception, sensory perception of the chemical stimulus, G protein-coupled receptor activity, and detection of stimulus. In the most significant MF, genes were associated with transmembrane signaling receptor activity. These responses occurred mainly in the membrane-bound cell organelles (Appendix A).

### 2.4. Defining the Disease-Residual Transcriptomic Profile (DRTP) 

To determine the disease-residual transcriptomic profile (DRTP), we examined the overlap between the DEGs of resolved vs. never-lesional skin from our study and the DEGs of the lesional vs. healthy skin from the available datasets at GEO for both epidermal and dermal compartments. Since we did not find any meaningful overlap between DEGs of the resolved vs. never-lesional dermis and DEGs of the lesional vs. healthy dermis, we focus on our results for the epidermal samples below.

Gene expression data from lesional and healthy epidermal skin samples were downloaded from the Gene Expression Omnibus (series matrix files of GSE68937 and GSE68923 datasets). A total of 4104 DEGs were identified in lesional epidermis vs. healthy epidermis. The psoriatic lesional-specific DEGs showed a 7-fold overrepresentation in resolved vs. never-lesional DEGs compared to non-DEGs of the same comparison (Fisher’s exact test *p* = 5 × 10^−53^). Overall, 102 genes overlapped between 476 DEGs of resolved vs. never-lesional and the 4104 DEGs of lesional vs. healthy in epidermal compartments. We found that 95% of these 102 overlapping genes had the same fold change direction in the two comparisons. The amount of change strongly correlated (Spearman’s rho: 0.75, *p* < 2.2 × 10^−16^); the data are shown in Figure 5.

Of these 102 overlapping genes, 67 were down-regulated, and 35 were up-regulated in the resolved epidermis (Appendix A). The five most down-regulated genes were **CACNA2D1**, **ODF3L1**, **WNT2**, **LIF**, and **TPPP**, whereas the five most up-regulated genes were **AKR1B10**, **FABP5**, **PYDC1**, **WNT5A**, and **TMEM52** (gene abbreviations are explained in Appendix A). Of note, the AKR1B10 gene was the top up-regulated overlapped gene between resolved and lesional epidermis. Combined transcriptomic analysis identified the AKR1B10 gene as the most differentially expressed gene in psoriatic lesions compared to healthy skin [41]. Microarray studies showed significantly higher expression of AKR1B10 (24-fold) in psoriatic lesions compared to healthy skin [39]. 

In addition, we determined the function of the overlapping genes through GO enrichment analysis. The results showed 54 significant BPs (*p* < 0.001 and FDR < 0.2). In the five most significant BPs, the genes were associated with the following terms: cellular response to stimulus, cellular response to transforming growth factor beta stimulus, response to transforming growth factor beta, positive regulation of the biological process, and Wnt signaling pathway involved in midbrain dopaminergic neuron differentiation (Appendix A). 

Then, the protein–protein interaction (PPI) networks of the 102 overlapping genes were generated using the STRING database. Based on the results, the largest network (NGFRAP1-PAMR1-TRAF6-ZC3H12A-RC3H1-PAN3-TREX2) was most likely associated with TNF signaling, whereas a cluster of WNT-signaling was also evident (Figure 6). 

The KEGG (Kyoto Encyclopedia of Genes and Genomes) pathway analysis was also performed on the 102 overlapping genes, which revealed the enrichment of the WNT and mTOR signaling pathways. Surprisingly, the overlapping genes were most significantly enriched in the basal cell carcinoma pathway (Table 2). Interestingly, the signaling pathways associated with breast cancer, gastric cancer, and hepatocellular carcinoma were also significantly up-regulated in the resolved epidermis. 

## 3. Discussion

Over the years, many studies have provided fundamental insights into the pathogenesis of psoriasis, and the knowledge has been translated into highly effective therapies. However, pathologic changes associated with lesion recurrence are only partially understood, making it difficult to develop practical strategies to prevent frustrating psoriasis flare-ups. A psoriasis relapse is defined by the appearance of new lesions and mainly by the recurrence of old lesions. The DRTP throughout psoriatic healed skin [14,15,16,17,18] and epidermal tissue-resident memory T cells (TRMs) [9,14,15,16] have been considered critical for the recurrence of old lesions. However, the role of resolved skin epidermal keratinocytes in disease recurrence is an unresolved issue. 

This is the first study to highlight the potential significance of epidermal keratinocytes in psoriasis local relapse by elucidating the DRTP and methylation/hydroxymethylation status in resolved epidermis. Our transcriptional results confirmed the distinction between transcriptomic profiles of never-lesional and resolved uninvolved skin in psoriasis. Among the DEGs in resolved epidermis compared with the never-lesional epidermis, the most down-regulated genes were functionally related to inflammation regulation (NEAT1-3) and genome integrity (SAMHD1). In addition, we found that genes involved in lipid metabolism (ELOVL3 and AKR1B10) and inflammation flare-up (C10orf99) were among the most up-regulated DEGs in resolved epidermis vs. never-lesional epidermis. Three of the most dysregulated genes, namely **SAMHD1**, **C10orf99**, and **AKR1B10**, were identified as likely to be important for lesion development in the resolved epidermis. We suggest these genes could confer a hypersensitivity of resolved epidermal keratinocytes to a secondary assault (Figure 7).

Since it was previously shown that **SAMHD1**-deficient fibroblasts from patients with Aicardi Goutières syndrome (AGS), a rare inflammatory disease, had elevated dNTP pools associated with chronic DNA damage and up-regulation of interferon (IFN)-stimulated genes [54], it is possible that SAMHD1-deficient resolved epidermal keratinocytes also have elevated dNTP levels compared to never-lesional cells, which may lead to latent and persistent DNA damage and inflammation. Thus, resolved epidermal keratinocytes may be more susceptible to genotoxic stress than never-lesional cells. 

**C10orf99** or GPR15L may act as a chemotactic ligand for GPR15 (G protein-coupled receptor 15). Transcriptomic studies and genomic analyses have shown that GPR15L is strongly up-regulated in the lesional skin of patients with psoriasis, atopic dermatitis, and in related animal models [26,40,55]. Single-cell RNA sequencing data showed that mainly epidermal keratinocytes express GPR15L in psoriatic lesional skin [56]. GPR15 is specifically expressed by effector/memory B cells, T cells, and regulatory T cells [50]. Whether the interaction between GPR15L and GPR15 plays a role in the initiation of psoriasis is unknown. However, higher expression of the chemotactic factor GPR15L in resolved epidermal keratinocytes may lead to retaining GPR15^+^ cells in resolved epidermis. 

The **AKR1B10** gene encodes a retinaldehyde reductase that plays a critical role in the retinoic acid (RA) or vitamin A metabolism and its activity causes decreased RA synthesis. Overexpression of AKR1B10 has been reported in patients with psoriasis [39], atopic dermatitis [57] and keloids [58]. This suggests that an imbalance in retinoic acid (RA) metabolism is a common feature of these relapsing–remitting inflammatory skin diseases, regardless of their pathological background. The lower intracellular level of RA is considered an inducing signal for cell proliferation and prohibiting for cell differentiation, both of which promote tumorigenesis in affected tissues [59]. In resolved epidermal keratinocytes, significant overexpression of AKR1B10 may decrease the level of RA by suppressing the RA synthesis pathway and increase cell survival and proliferation by activating the retinal–retinol pathway. On the other hand, the lower expression of AKR1B10 gene in never-lesional epidermal keratinocytes may increase the level of RA and induce keratinocytes differentiation while decreasing cell survival. However, the abovementioned hypotheses need to be investigated and are open to future research on psoriasis local relapse. 

In addition, we defined the DRTP as a set of 102 expressed genes that overlapped between the DEGs of resolved vs. never-lesional and the DEGs of lesional vs. healthy skin. Remarkably, we found that the **AKR1B10** transcript was not only among the most up-regulated DEGs in the resolved epidermis compared to the never-lesional epidermis, but was also the most up-regulated transcript overlapping between resolved and lesional epidermis. This finding suggests that the retinoic acid signaling pathway plays an essential role in the local recurrence of psoriatic lesions. Indeed, **acitretin**, a widely used drug for psoriasis, acts through the retinoic acid signaling pathway and affects keratinocytes proliferation and differentiation [60]. Since AKR1B10 is also a druggable target [61], the repurposing of the already known AKR1B10 inhibitors could be of translational importance. 

We also found that **Wnt5a**, a negative regulator of epidermal keratinocyte differentiation, remained strongly up-regulated in resolved epidermal keratinocytes. This is consistent with previous findings that recognized the Wnt5a gene as DRTP in psoriatic healed skin after successful etanercept therapy [14]. Our STRING and KEGG pathway analyses also revealed the up-regulation of genes involved in the Wnt pathway. Overall, these data support the hypothesis that the WNT signaling pathway may contribute to the recurrence of psoriatic lesions. Furthermore, TNF and mTOR signaling pathways, which play a role in the pathogenesis of psoriasis, were among the major disease-residual pathways identified in our STRING and KEGG pathway analyses. 

We also demonstrated a clear difference in the 5-mC and 5-hmc general pattern between psoriatic never-lesional and resolved, uninvolved skin. In resolved epidermis, in addition to 5-mC and TET3 mRNA expression levels compared with never lesional epidermis, we also found greatly decreased 5-hmC contents. Loss of 5-hmC has been reported in different solid tumors [62] and immune-mediated skin disorders, such as psoriasis [28]. Since 5-mC is the only substrate to produce 5-hmC in vivo [63], the loss of 5-hmC in resolved epidermal keratinocytes could result from a global decrease in 5-mC content. On the other hand, TET enzyme dysfunction or decreased levels of it may also result in reduced 5-hmC generation in cells. Hence, low 5-mC content and TET3 enzyme deficiency within the resolved epidermis possibly account for the loss of 5-hmC in resolved epidermal keratinocytes (Figure 8). Interestingly, perturbation of TET-5-hmC pathway in the epidermis of psoriatic lesions has been already reported. It has been suggested that this is related to the loss of the self-renewal capacity of basal keratinocytes that become FABP5-expressing transient amplifying cells (TACs) [28]. In our study, FABP5 remained up-regulated (2.09-fold) in the hypo-hydroxymethylated resolved epidermis (Table 1). These results and the disrupted Wnt5a signaling pathway in the resolved epidermis suggest that epidermal keratinocytes in healed/resolved skin harbor cell differentiation defects at the transcriptomic and epigenetic levels. As mentioned previously, we also observed that DNA was hypo-methylated in resolved epidermal keratinocytes compared to never-lesional cells. The hypo-methylated DNA profile, defined as a decreased 5-mC content of the genome, is a hallmark of several cancers [64] and autoimmune diseases [65,66,67,68,69]. It has been reported that auto-reactive T cells in systemic lupus erythematosus (SLE) [65,66], synovial fibroblasts in rheumatoid arthritis (RA) [67], and cells in the white matter of multiple sclerosis (MS) scars are globally hypo-methylated [68,69]. DNA methylation profiling in psoriasis patients showed intermediate methylation differences in psoriatic uninvolved skin compared to healthy and psoriatic lesional skin [26]. In summary, our findings suggest that disease-residual epigenomic and transcriptomic profiles are still present in resolved epidermal keratinocytes after successful therapy. 

## 4. Materials and Methods

### 4.1. Psoriatic Skin Samples Collection and Ethics 

Seven volunteer patients with moderate-to-severe plaque-type psoriasis, aged >25 years, were enrolled in our study. The disease severity was evaluated using the Psoriasis Area and Severity Index (PASI) scoring system. The PASI is a widely used and gold standard measurement tool that grades the severity of psoriatic lesions and assesses the treatment response of psoriasis patients [70]. This study recruited patients who matched the criteria irrespective of sex and therapeutic regimen. Our inclusion criteria were to have patients with moderate-to-severe (PASI above 15) plaque-type psoriasis before initiating systemic therapy and being on systemic therapy for at least 1 year before taking their skin samples. Psoriatic patients’ detailed information and experimental techniques applied to their samples are available in Appendix A. Psoriatic tissue collection was obtained after written informed consent according to the rules of the Helsinki Declaration. Protocols for obtaining patient biopsies were approved by the Regional and Institutional Research Ethics Committee (PSO-CELL-01, 90/2021, 4969, 26 April 2021, Szeged, Hungary; HCEMM-001, 10/2020, 4702, 20 January 2020, Szeged, Hungary) for the protection of human subjects. Full-thickness 6 mm paired never-lesional and resolved skin punch biopsies (PBs) were taken under aseptic conditions with local anesthesia from psoriasis patients. At the time of sampling, their resolved skin had been resolved for at least 6 months. Both the clinician and the patient determined the resolved skin, and the lesional skin was obvious to detect. 

### 4.2. Co-Localization of 5-mC and 5-hmC Epigenetic Marks in Psoriatic Never-Lesional and Resolved Skin Sections

First, we set up the protocol for equilibrium binding of 5-mC and 5-hmC antibodies to the target antigens in frozen paraformaldehyde-fixed healthy skin sections (Appendix B). Shortly, never-lesional and resolved skin PBs were immediately fixed in a freshly made 4% paraformaldehyde (PFA 4%) solution in PBS (containing 0.01% TritonX-100) for 6 h on a rocker at room temperature (RT). Then the samples were washed once with PBS for 5 min. To minimize freezing-induced damages, the samples were infused gradually with sucrose by placing them in 10% and 20% sucrose in PBS (1 h each by rocking at RT) and 30% sucrose (overnight at 4 °C) [71]. The tissues were subsequently frozen in the cryogenic matrix (Thermo Fisher Scientific, Waltham, MA, USA) using liquid nitrogen, and 12 μm sections were cut and stored at −20 °C until further processing. The sections were washed 2 × 15 min in PBS to remove the infused sucrose and were incubated for 10 min in 0.1% Triton X-100 in PBS at RT. For antigen retrieval, sections were incubated for 1 h with freshly made 2 N hydrochloric acid (HCl) in PBS at RT. After DNA denaturation, sections were neutralized using 0.1 M Tris-HCl (pH 8.3) for 10 min. For blocking, the sections were incubated in blocking solution ((1% normal goat serum, 1% bovine serum albumin (Sigma-Aldrich, St. Louis, MO, USA) in PBS)) for 1 h, at RT in a humidified chamber. Samples were incubated with the following primary antibodies: polyclonal rabbit anti-human 5-hmC (1:1000, Active motif, Carlsbad, CA, USA, Cat No. 39769), mouse monoclonal anti-human 5-mC (1:500, Epigentek, East Farmingdale, NY, USA, Cat No. A-1014) for over-night at 4 °C. Purified mouse IgG1 κ isotype (Biolegend, San Diego, CA, USA, Cat No. 400102) was used for 5-mC isotypic control. As secondary antibodies, AlexaFluor 546-conjugated anti-rabbit IgG and AlexaFluor 647-conjugated anti-mouse IgG (Life Technologies, Carlsbad, CA) were used (both 1:500). We visualized the nuclei with DAPI (Sigma-Aldrich, St. Louis, MO, USA) staining.

### 4.3. Microscopy

Olympus Fluoview FV1000 (Olympus Life Science Europa GmbH, Hamburg, Germany) confocal laser scanning microscope was used for fluorescence imaging following immunolocalization. Dry (10×, 20×) and oil immersion (PLANAPO 40× and 60×) objectives were used during imaging. The default DAPI and AlexaFluor dye combination setup of the microscope was used to capture images. Transmitted light detector of the microscopes were used for bright field images and merged onto DAPI images using Olympus Fluoview software. As regards Figure 3, identical microscope configuration is used for all intensity comparison images where ICA look up table of FIJI open software was employed after identical enhancing of brightness values of never-lesional and resolved pairs [72].

### 4.4. RNA Extraction and Real-Time RT-PCR

The skin PBs were washed in cold saline containing antibiotic/antimycotic (AB/AM) to remove the blood. Subsequently, the skin PBs were incubated in Dispase II solution (neutral protease, grade II, 2 U/mL, Roche Diagnostics, Basel, Switzerland) for 3 h at 37 °C and the epidermis was then carefully peeled from the dermis [73]. Following that, the epidermal and dermal compartments were immediately submerged in RNAlater (Invitrogen, Waltham, MA, USA, Cat No. AM7020) for storage until further processing for RNA extraction. Only the Ustekinumab-treated patients were included for full-length transcriptome sequencing (Appendix A). Epidermal and dermal samples were mechanically homogenized using the Ultra Turrax T8 homogenizer (IKA-WERKE, Staufen, Germany). Total RNA was extracted using TRI-Reagent (Molecular Research Center; Cincinnati, OH, USA). The purity and concentration of RNA samples were determined via Nanodrop (Colibri Mikrovolüm Spektrometre, Berthold, Germany). The qualified RNA samples were used for cDNA synthesis using UltraScript cDNA Synthesis Kit (Thermofisher, Waltham, MA, USA). Changes in mRNA expression were detected with real-time RT-PCR using the FAM dye-labeled TaqMan^TM^ probes (Thermofisher, Waltham, MA, USA). TaqMan^TM^ probes used in our study are listed in Appendix A. Real-time RT-PCR experiments were carried out using the qPCRBIO Probe Mix Lo-ROX (PCR Biosystem Ltd., London, UK) on a C1000 Touch Thermal Cycler (Bio-Rad Laboratories, Hercules, CA, USA). All reactions were duplicated, and the data were normalized to the 18S ribosomal RNA gene. Relative mRNA expression was calculated using the ΔΔCt method. Data from never-lesional and resolved epidermis were compared using an unpaired two-tailed *t*-test. Differences were considered significant when *p* < 0.05. 

### 4.5. High-Throughput mRNA Sequencing

A high-throughput mRNA sequencing analysis was performed on the Illumina sequencing platform to obtain global transcriptome data. Total RNA sample quality was checked on Agilent BioAnalyzer using the Eukaryotic Total RNA Nano Kit according to the manufacturer’s protocol. Samples with RNA integrity number (RIN) value > 7 were chosen for the library preparation process. RNA sequencing libraries were prepared from total RNA using an Ultra II RNA Sample Prep kit (New England BioLabs, Ipswich, MA, USA) according to the manufacturer’s protocol. Briefly, poly-A RNAs were captured by oligo-dT conjugated magnetic beads and the mRNAs were then eluted and fragmented at 94 °C. First-strand cDNA was generated through random priming reverse transcription, and double-stranded cDNA was developed after the second-strand synthesis step. After repairing ends, A-tailing, and adapter ligation steps, adapter-ligated fragments were amplified in enrichment PCR; finally, sequencing libraries were generated. Sequencing runs were executed on Illumina NextSeq 500 instrument using single-end 75-cycle sequencing.

#### 4.5.1. RNA Sequencing Data Analysis

Raw sequencing data (fastq) were aligned to human reference genome version GRCh38 using the HISAT2 algorithm, and BAM files were generated. Downstream analysis was performed using StrandNGS software (www.strand-ngs.com, The access date was 8 May 2022). BAM files were imported into the software, and the DESeq algorithm was used for normalization. Moderated *t*-test was used to determine DEGs in resolved versus never-lesional skin. Genes with *p* < 0.05 were considered DEGs. Raw sequencing data is available in the NCBI under the BioProject ID: PRJNA938026 (https://www.ncbi.nlm.nih.gov/sra/PRJNA938026, accessed on 20 December 2022). 

#### 4.5.2. Pathway Analyses

Cytoscape v3.4 software with ClueGo v2.3.5. application was used for identifying over-represented gene ontology (GO) terms. A two-sided hypergeometric test with Benjamini–Hochberg FDR correction was performed using the list of DEGs and the GO biological process database. 

### 4.6. Examining the Overlap between Resolved vs. Never-Lesional DEGs and Lesional vs. Healthy DEGs

Gene expression data of psoriatic and healthy epidermis samples were downloaded from Gene Expression Omnibus (series matrix files of GSE68937 and GSE68923 datasets). GSE68937 and GSE68923 contained the microarray results from two models, including three lesional and two healthy epidermis samples in sets one and three lesional and three healthy epidermis samples in set 2 [74]. Differentially expressed genes between lesional and healthy epidermal samples were identified using Wilcoxon’s rank-sum test. *p* values were adjusted according to Benjamini and Hochberg [75], and genes with a false discovery rate lower than 0.1 (FDR < 0.1) were classified as DEGs. IDs were collated using data from the Human Gene Nomenclature Organization (HUGO) website. Fold change values were calculated by dividing the median gene expression values in psoriatic with the ones in healthy samples. The overlapping gene function was identified with a GO enrichment analysis using the GOrilla software [76,77]. We classified overlapping DEGs as target genes and all DEGs identified in resolved as background genes. Biological processes with *p* < 0.001 and FDR < 0.2 were considered significant. KEGG pathway enrichment analysis was carried out with WebGestalt online tool. The classification of genes was the same as for the study with GOrilla. Pathways with an FDR < 0.2 were considered significantly enriched. PPI networks were generated with STRING v11.5 using default parameters and at least medium confidence interactions. R program (version R 3.6.3) was used for statistical analyses. 

### 4.7. Statistical Analysis 

Statistical analyses and the graphs for the real-time RT-PCR results were performed using GraphPad Prism 8.0.2 (GraphPad Software, San Diego, CA, USA). 

## Figures and Tables

**Figure 1 ijms-24-04556-f001:**
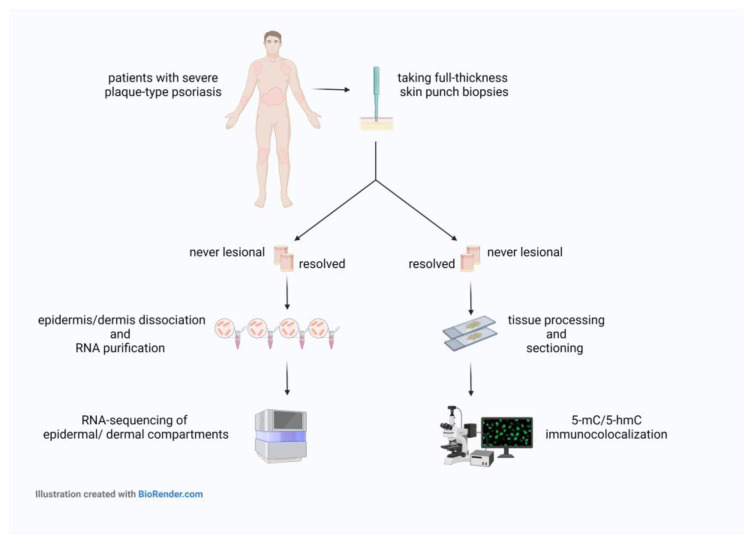
Schematic view of the experimental workflow. Abbreviations: 5-mC, 5-methylcytosine; 5-hmC, 5-hydroxymethylcytosine.

**Figure 2 ijms-24-04556-f002:**
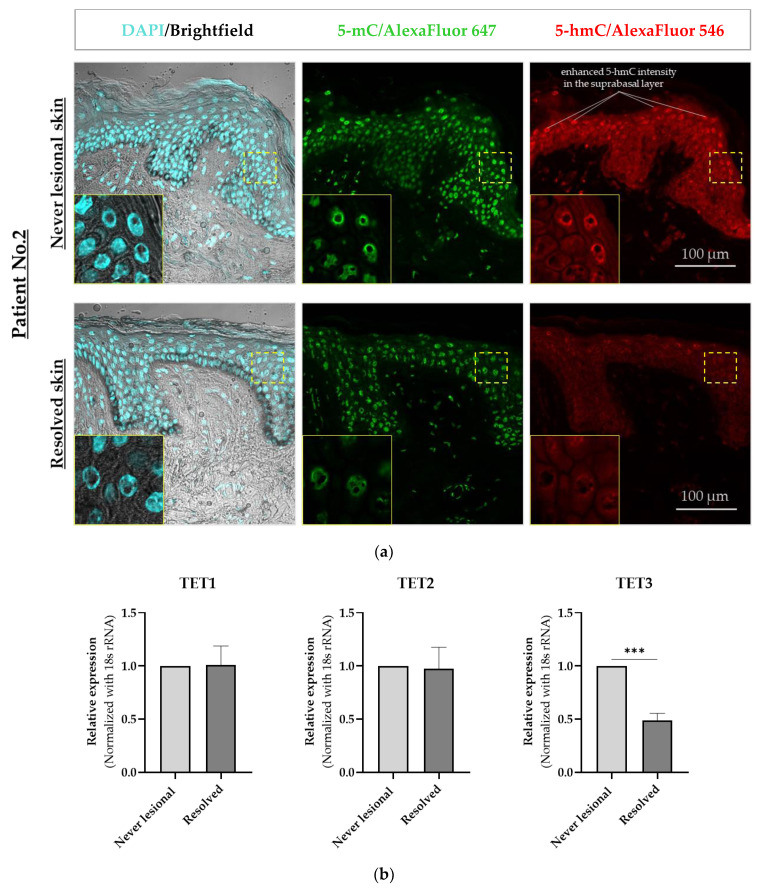
Confocal microscopy immunofluorescence analysis of 5-mC and 5-hmC in never-lesional vs. resolved skin. (**a**) Representative images are shown (*n* = 4). Colocalization immunofluorescence staining of 5-mC (green) and 5-hmC (red). Closeup images show yellow-dotted regions at higher magnification. Scale bars are 100 μm. (**b**) Real-time RT-PCR of TET1, TET2, and TET3 in resolved epidermis vs. never-lesional epidermis. An unpaired two-tailed *t*-test was performed, *** *p* < 0.001. Data = mean ± SEM of biological replicates (*n* = 3). Abbreviations: 5-mC, 5-methylcytosine; 5-hmC, 5-hydroxymethylcytosine; TET1, ten-eleven translocation (TET)1; TET2, ten-eleven translocation (TET)2; TET3, ten-eleven translocation (TET)3.

**Figure 3 ijms-24-04556-f003:**
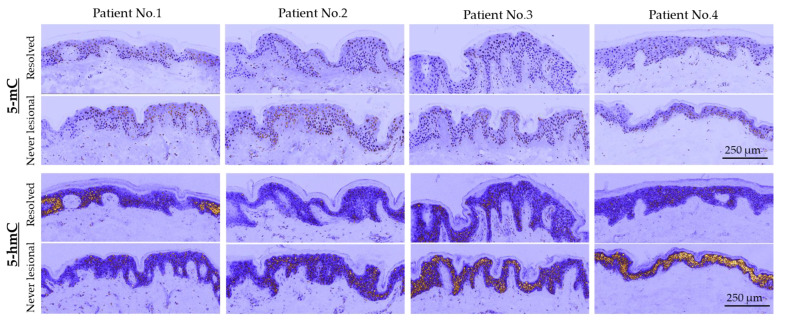
False-colored (pseudocolored) images of skin sections with a low magnification (20×) objective. All never-lesional and resolved pairs are identically brightness-adjusted. Scale bars are 250 μm. Abbreviations: 5-mC, 5-methylcytosine; 5-hmC, 5-hydroxymethylcytosine.

**Figure 4 ijms-24-04556-f004:**
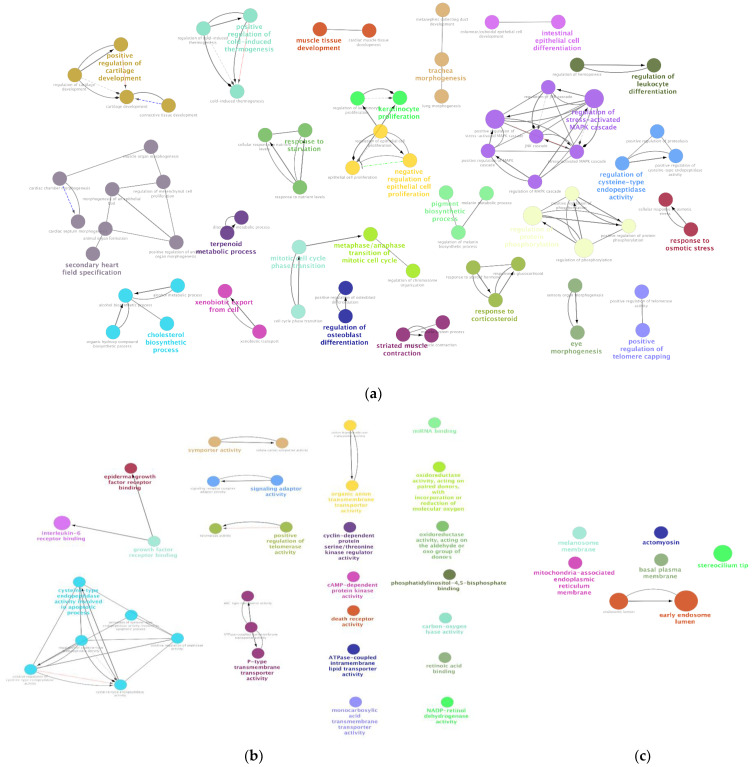
GO enrichment analysis of resolved vs. never-lesional epidermis DEGs. Enrichment shows only significant (**a**) BPs, (**b**) MFs, and (**c**) CCs with *p* < 0.05. Node color represents the specific functional class, biological processes, and cellular components involved in the enrichment analysis of the identified DEGs. Bold fonts indicate the major BPs, MFs, or CCs that define the names of each group. Abbreviations: DEG, differentially expressed genes; BPs, biological processes; MFs, molecular functions; CCs, cellular components.

**Figure 5 ijms-24-04556-f005:**
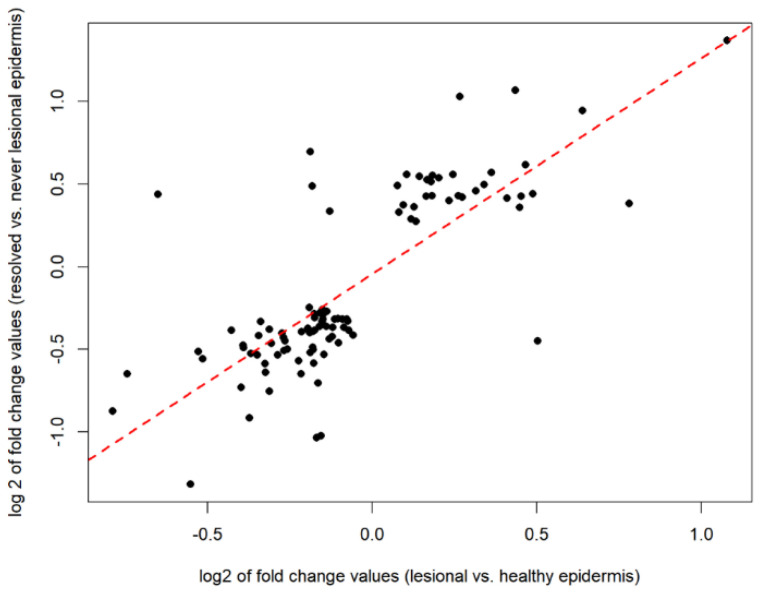
The Spearman’s rank correlation coefficient analysis was performed on 102 overlapping genes between 476 DEGs of the resolved vs. never-lesional and the 4104 DEGs of the lesional vs. healthy in epidermal compartments. The amount of change strongly correlated. Abbreviations: DEGs, differentially expressed genes.

**Figure 6 ijms-24-04556-f006:**
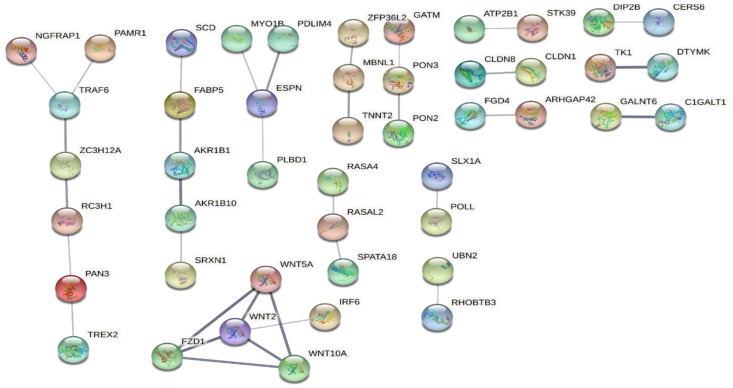
PPI networks of the 102 overlapping genes between resolved vs. never-lesional DEGs and lesional vs. healthy DEGs were generated using the STRING database. The figure shows the result ordered by network size. Abbreviations: PPI, protein–protein interaction; DEGs, differentially expressed genes.

**Figure 7 ijms-24-04556-f007:**
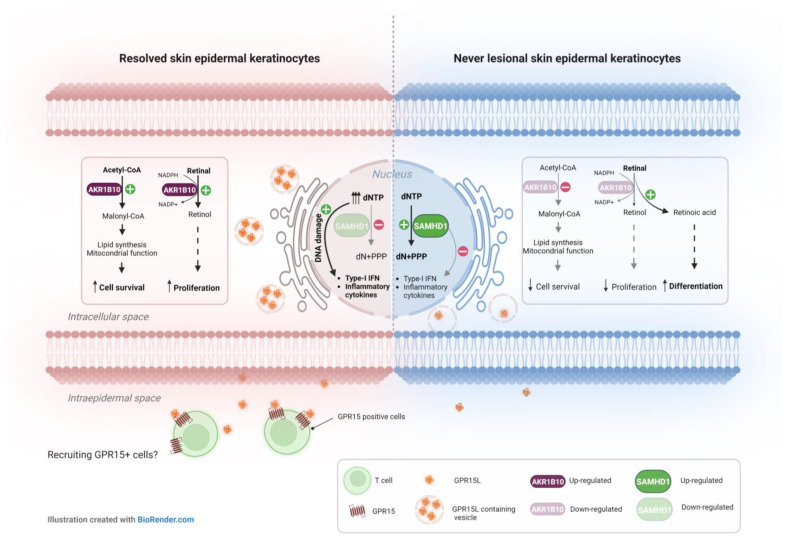
Mechanism of action of the three DEGs in resolved vs. never-lesional epidermis identified as likely to be important for lesion development in resolved epidermis. Significantly lower levels of the **SAMHD1** gene in resolved epidermal keratinocytes might be accompanied by abnormal dNTP accumulation, which can induce DNA damage. It is established that damaged DNA species can trigger type-I IFN and innate immune responses. At the same time, SAMHD1 higher expression level in never-lesional epidermal keratinocytes may keep DNA integrity more efficiently. Strong up-regulation of **GPR15L** in resolved epidermal keratinocytes may lead to the recruitment of GPR15+ cells, such as T-cells, into the epidermal compartment. In resolved epidermal keratinocytes, the strong up-regulation of **AKR1B10** gene may result in reduced RA and consequently increased keratinocytes survival and proliferation. However, the lower levels of AKR1B10 gene in never-lesional epidermal keratinocytes can activate the retinal–retinoic acid pathway and induce keratinocyte differentiation, while decreasing cell survival. Abbreviations: DEGs, differentially expressed genes; CoA, Coenzyme A; NADPH, nicotinamide adenine dinucleotide phosphate; dNTP, deoxynucleotide triphosphates; PPP, triphosphate; dN, deoxyribonucleoside.

**Figure 8 ijms-24-04556-f008:**
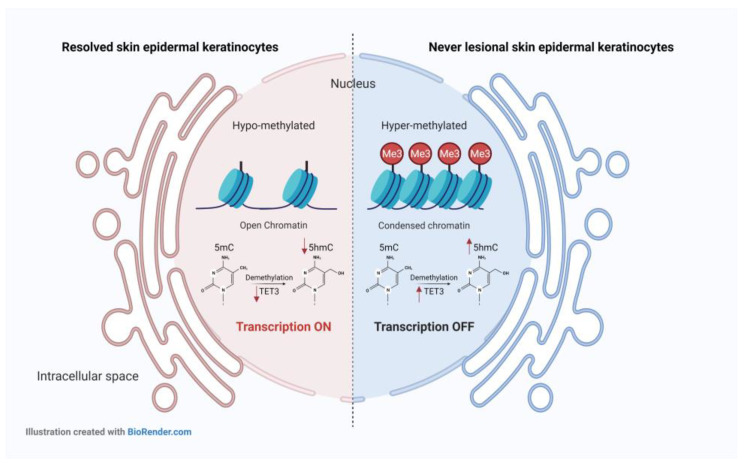
The DNA methylation/hydroxymethylation status of resolved vs. never-lesional epidermal keratinocytes. The DNA was hypo-methylated and hypo-hydroxymethylated in resolved vs. never-lesional epidermal keratinocytes. Hypo-methylated DNA is generally associated with open chromatin and transcription activation. Hence, the transcription might be more active in hypo-methylated DNA of resolved epidermal keratinocytes than never-lesional ones. Furthermore, hypo-methylated DNA in the resolved epidermis can reduce 5-hmC DNA levels. At the same time, the lower mRNA expression of TET3 enzyme might also be another reason for the loss of 5-hmC in the resolved epidermis. Abbreviations: 5-mC, 5-methylcytosine; 5-hmC, 5-hydroxymethylcytosine; TET3, ten-eleven translocation (TET)3.

**Table 1 ijms-24-04556-t001:** The 25 most down-regulated and up-regulated DEGs in epidermal and dermal compartments. The 25 most down-regulated and up-regulated DEGs in resolved vs. never-lesional were sorted according to their FC. Abbreviations: DEG, differentially expressed genes; FC, fold of change.

The 25 Most Down-Regulated and Up-Regulated Resolved vs. Never-Lesional DEGs in Epidermal and Dermal Compartments									
Epidermis (*n* = 4)					Dermis (*n* = 3)				
	Down-Regulated		Up-Regulated			Down-Regulated		Up-Regulated	
Number	Gene Symbol	FC	Gene Symbol	FC	Number	Gene Symbol	FC	Gene Symbol	FC
1	NEAT1_3	−4.66	STAC2	5.35	1	SPRR4	−32.61	IGHV3-7	12.68
2	SAMHD1	−2.57	AQP5	4.38	2	ATP12A	−25.92	IGHV3-33	10.90
3	HOXB2	−2.50	FAM25C	3.36	3	CST6	−17.10	IGHJ4	10.56
4	CACNA2D1	−2.49	ELOVL3	3.31	4	SPRR1A	−17.00	IGKV3-20	7.39
5	FHL1	−2.22	C10orf99	2.75	5	CRCT1	−14.41	IGKV3-15	7.01
6	RPL24P7	−2.14	AKR1B10	2.57	6	MSMB	−12.93	IGLV1-44	6.40
7	SNORA9	−2.11	FAM25G	2.42	7	KRT34	−10.56	POU6F2	6.10
8	CYP4B1	−2.05	CRNN	2.41	8	GJB4	−10.54	IGKV1D-12	5.72
9	ODF3L1	−2.04	RPL31P63	2.28	9	RNF222	−9.99	LINC00619	5.61
10	WNT2	−2.03	FABP5P2	2.28	10	CTSV	−9.46	ADH4	5.57
11	KLHL11	−1.94	FABP5P10	2.26	11	KRTAP3-2	−9.37	IGKV1-17	5.03
12	TMEM256	−1.93	LGR6	2.20	12	KRT33A	−8.82	IGKV1-12	4.87
13	PELI2	−1.88	IL1F10	2.19	13	PLA2G2F	−8.56	IGHV2-70	4.69
14	WDR82P2	−1.88	FABP5P1	2.12	14	SLC15A1	−8.40	ITIH1	4.49
15	LIF	−1.88	FABP5	2.09	15	KRT31	−8.30	IGKV1-16	4.38
16	TPPP	−1.83	WISP3	2.09	16	RPP21	−8.29	ADAM20P1	4.33
17	RCAN2	−1.77	FABP5P7	2.09	17	TRIM39-RPP21	−7.85	SAA2	4.27
18	PKIB	−1.74	NPM1P25	2.07	18	SPRR1B	−7.48	IGHV1-18	4.16
19	MIR23A	−1.71	MRPS10P1	2.07	19	B4GALNT2	−7.48	RHPN1-AS1	4.10
20	KLF9	−1.68	PYDC1	2.03	20	CASP1P2	−7.15	ANP32BP3	3.99
21	LRRC8C	−1.68	IGFBP2	2.00	21	BPIFC	−6.94	RHOXF2	3.96
22	SLC47A1	−1.67	RPS26P39	2.00	22	KRT38	−6.89	IGHV1-69	3.93
23	CNTNAP3P2	−1.66	ENTPD3	1.98	23	LIPK	−6.87	OLAH	3.89
24	IKZF2	−1.66	KRT2	1.94	24	CARD18	−6.43	IGLV1-47	3.83
25	WDR45BP1	−1.66	WNT5A	1.92	25	PCSK1N	−6.14	MT3	3.79

**Table 2 ijms-24-04556-t002:** KEGG pathway enrichment analysis was performed on the 102 overlapping genes between DEGs of the resolved epidermis vs. never-lesional epidermis and DEGs of the lesional epidermis vs. healthy epidermis. The results showed eight significantly enriched pathways (*p* < 0.001 and FDR < 0.2). Abbreviations: KEGG, Kyoto Encyclopedia of Genes and Genomes; DEGs, differentially expressed genes.

The KEGG Enrichment Analysis of the 102 Overlapping Genes between Resolved and Lesional Epidermis							
	Gene Set	Description	Size	Expect	Ratio	*p* Value	FDR
1	hsa05217	Basal cell carcinoma	59	0.38756	12.901	3.78 × 10^−5^	0.0122
2	hsa04550	Signaling pathways regulating pluripotency of stem cells	132	0.86709	5.7664	1.63 × 10^−3^	0.1296
3	hsa04310	WNT signaling pathway	140	0.91964	5.4369	2.12 × 10^−3^	0.1296
4	hsa05224	Breast cancer	142	0.93277	5.3604	2.25 × 10^−3^	0.1296
5	hsa05226	Gastric cancer	142	0.93277	5.3604	2.25 × 10^−3^	0.1296
6	hsa04150	mTOR signaling pathway	144	0.94591	5.2859	2.39 × 10^−3^	0.1296
7	hsa04916	Melanogenesis	95	0.62404	6.4099	3.36 × 10^−3^	0.1534
8	hsa05225	Hepatocellular carcinoma	160	1.051	4.7573	3.78 × 10^−3^	0.1534

## Data Availability

Data are presented in the manuscript and are available upon request from the corresponding author.

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
