# Peer review of "Psoriatic Resolved Skin Epidermal Keratinocytes Retain Disease-Residual Transcriptomic and Epigenomic Profiles"

_ijms, 2023, doi:10.3390/ijms24054556_

Round 1
Reviewer 1 Report
I read with great interest this manuscript titled "Psoriatic resolved skin epidermal keratinocytes retain disease 2 residual transcriptomic and epigenomic profiles" by Ghaffarinia et colleagues.
The methodology is solid and offers an interesting perspective by integrating transcriptomics and epigenetics.
I have only some minor suggetsions:
Please add a more updated referenc efor the prevalence of psoriasis (i.e. GBD psoriasis from Frontieris)
Please discuss how diet and pollution may moderate epigenetics in the discussion
Please add the Supplemntary figure S2 program to make the picture
Please add the database you produced for more transparence
Author Response
Dear Reviewer 1,
Please see the attachment.

Reviewer 2 Report
- How did you differentiate resolved skin from lesional skin? The PASI socres of the patients are quite high, which means that the patients are suffering from active psoriasis, and mild psoriatic skin could look like 'resolved skin'. It would be better to provide how long did the 'resolved skin' maintain the - status of 'resolved'.
- How can you say that these epigenetic changes are the 'cause' of site-specific local relapse, not the 'result' of the previous psoriasis event?
Author Response
Dear Reviewer 2,
Please see the attachment.

Reviewer 3 Report
An interesting study, that will be eligible to be published after revisions:
Lines 51-54, you need some references, such as: doi: 10.1111/dth.13185. and doi: 10.3390/pharmaceutics14020294.
The critical points of the study can be identified in the sample's smallness and in the observation time's short duration. The study is a single-center observational study; therefore, the samples examined come from a single care setting; this aspect reduces the clinical applicability of the study results. Historically, psoriasis was viewed as a complex immune-mediated disease in which T lymphocytes, dendritic cells, and cytokines (interleukin [IL] 23, IL-17, and tumor necrosis factor [TNF]) play a central role; however, there are few reports on biomarkers associated with psoriasis in saliva. The markers examined are too nonspecific, as they are altered in other stress-related pathologies. Exclusion criteria exclude obesity, psychiatric illness, and all other autoimmune conditions except lupus. The endpoint considered would be more interesting, with a larger sample, longer observation times, and more stringent inclusion and exclusion criteria.
However i find the study publishable after revisions
Author Response
Dear Reviewer 3,
Please see the attachment.

Reviewer 4 Report
The residual transcriptome of disease (DRTP) in the resident memory T (TRM) cells in the skin and epidermis of psoriasis has been proposed as the basis for the recurrence of old lesions. This study used epigenetic methods to sequence RNA in the epidermis and dermis of psoriasis patients. The results showed that the mRNA expression of 5-mC, 5-mC and Teneleven translocation (TET) 3 enzyme decreased. Therefore, it is concluded that DRTP of keratinocytes may contribute to the cause of site-specific local recurrence. This study has certain significance for understanding the recurrence mechanism of psoriasis. However, the following questions need to be answered by the author.
1) How to determine the severity of psoriasis patients? What are the specific indicators?
2) How are dermal and epidermal samples obtained? Details should be presented in the methods.
3) What are the criteria for selecting patients?
4) Are there differences between male and female? Is there only one woman in the test?
Why are there no IM experiment using female samples?
5) Why did the author not test the enrichment of WNT and mTOR signal pathways?
6) Can the author detect WNT and mTOR signal pathway proteins in samples?
Author Response
Dear Reviewer 4,
Please see the attachment.
